# Efficient association mapping from k-mers— An application in finding sex-specific sequences

**Zakaria Mehrab**[1,2], **Jaiaid Mobin**[1], **Ibrahim Asadullah Tahmid**[1], **Atif Rahman**[1] *

**1** Department of Computer Science and Engineering, Bangladesh University of Engineering and Technology, Dhaka, Bangladesh, **2** Department of Computer Science and Engineering, United International University, Dhaka, Bangladesh

* atif@cse.buet.ac.bd

**Data Availability Statement:** The E. Coli ampicillin resistance dataset can be downloaded from the Sequence Read Archive (SRA Accession No. SRR3050845 - SRR3051085). The scripts, source code, and user manual for HAWK is available at

## Abstract

Genome wide association studies (GWAS) attempt to map genotypes to phenotypes in organisms. This is typically performed by genotyping individuals using microarray or by aligning whole genome sequencing reads to a reference genome. Both approaches require knowledge of a reference genome which hinders their application to organisms with no or incomplete reference genomes. This caveat can be removed by using alignment-free association mapping methods based on k-mers from sequencing reads. Here we present an improved implementation of an alignment free association mapping method. The new implementation is faster and includes additional features to make it more flexible than the original implementation. We have tested our implementation on an *E. Coli* ampicillin resistance dataset and observe improvement in execution time over the original implementation while maintaining accuracy in results. We also demonstrate that the method can be applied to find sex specific sequences.

## Introduction

Association mapping is the process of associating phenotypes with genotypes. In genome wide association studies (GWAS), individuals are typically genotyped using microarrays or by aligning sequencing reads from individuals to a reference genome. However, both these approaches require a reference genome of the organism which makes them inappropriate for association mapping in non-model organisms with incomplete reference genomes or none at all.

To address this issue, alignment free approaches for association mapping have been explored. A number of methods have been developed to perform association studies in bacterial genomes that do not require aligning reads to reference genomes [1–4]. The high plasticity in bacterial genomes means structural variants and even large genomic segments in various strains are missing in the reference genomes which makes application of reference based methods difficult. However, these methods do not scale to organisms with large genomes, and as many of them have incomplete reference genomes, there were challenges in association mapping in these organisms. To overcome this, Rahman et al. [5] and Voichek et al. [6] presented

**Funding:** The authors received no specific funding for this work.

**Competing interests:** The authors have declared that no competing interests exist.

methods for mapping associations in large genomes, to categorical phenotypes and to both categorical and quantitative phenotypes, respectively. The methods are primarily based on finding k-mers i.e. contiguous sequences of length k in sequenced reads and identifying k-mers associated with the phenotype.

In the association mapping tool named HAWK developed by Rahman et al. [5], frequencies of k-mers are analyzed to find k-mers associated with a phenotype and then they are assembled to form the associated sequences. First, they count k-mers in reads from each individual using Jellyfish [7]. Second, using likelihood ratio test, they find k-mers with significantly different counts in case and control samples. Next, population structure is determined from k-mer counts using Eigenstrat [8, 9]. After that, associations to k-mers after correcting for population structure are determined. Finally, the k-mers found associated may be assembled to get a sequence for each associated loci.

Here we re-implement HAWK with the goal to reduce its execution time and make it more convenient for users. We have re-implemented the step for finding associated k-mers after population structure correction using C++, which was previously implemented in R. We have also extended support for Jellyfish 2 and implemented Benjamini–Hochberg procedure [10], which can be used to correct for multiple tests when the study is underpowered for Bonferroni correction. We have tested our implementation with a dataset on *E.coli* ampicillin resistance and have compared its output with the output of the original implementation. We have also analyzed the execution times of the two implementations. Our implementation is faster and more flexible to run compared to the original implementation while producing results similar to the original one.

Finally, we show that our method can be used to find sequences in the sex chromosomes. We apply our method to sequencing data from two populations in the 1000 genomes dataset [11], labeling males and females as cases and controls. We find that the k-mers determined by HAWK cover the entire sequenced regions in X and Y chromosomes. It is worth noting that other reference free methods for association mapping mentioned above are based on presence and absence of k-mers, and hence are not suitable for finding sequences in sex chromosomes present in both sexes e.g. the human chromosome X.

## Implementation

Here we summarize the improvements and new features we have added. Fig 1 shows the workflow for HAWK highlighting the additions and modifications. Results supporting the improvements is presented in the following section.

### Re-implementation of correction for population structure

Population stratification is a known confounder in association studies. Without correcting for this confounding factor, one may falsely associate non-significant genotypes with phenotypes. In the HAWK pipeline, population structure was estimated using Eigenstrat [8, 9]. Eigenstrat performs a principal component analysis (PCA) on the presence or absence status in each sample of a randomly selected set of k-mers. Then the population structure is represented by the projections of the data points along the principal components. By default, first two principal components are used to denote population stratification but users have the option to choose up to first ten of them. Subsequently, p-values were adjusted for population structure and other confounders i.e. the associations between the k-mers and the phenotype were adjusted for confounders and p-values were re-estimated using the glm function (for fitting logistic regression models) and the ANOVA function (for testing the goodness of fit) in R.

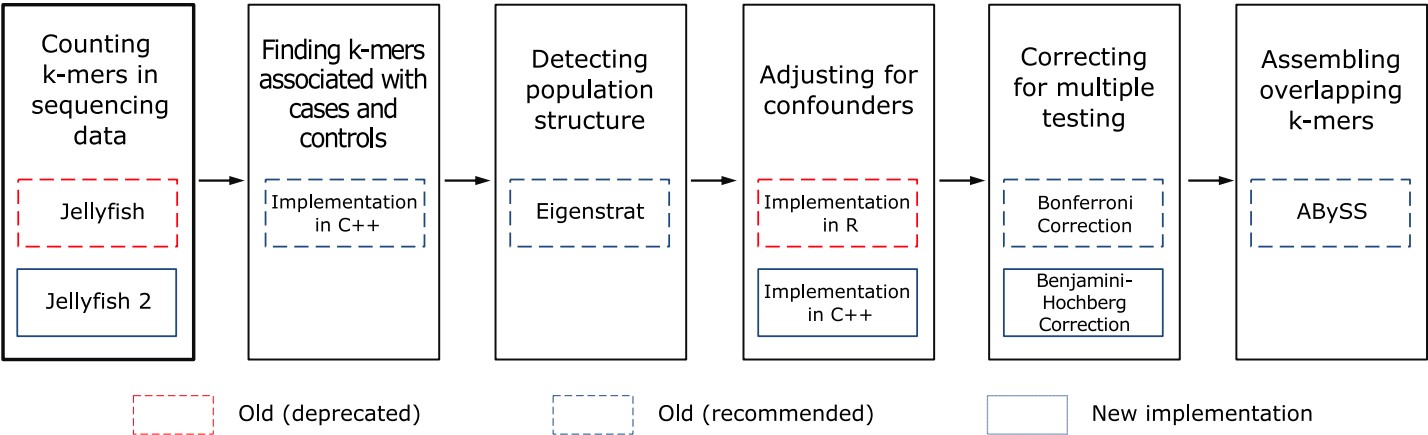

**Fig 1. The workflow for the reference free association mapping method HAWK with the new features highlighted.** The re-implemented and newly added features are shown within solid blue boxes. Dashed red boxes and dashed blue boxes indicate old implementations that have been deprecated and are still recommended respectively.

Here, we re-implement this process in C++ and thereby, improve the performance of the pipeline. R uses the IRLS (Iteratively Re-weighted Least Square) method to fit the model [12]. Therefore in our implementation of the glm function in C++, we also used IRLS for fitting the model. Iteratively re-weighted least squares for finding the MLE (Maximum Likelihood Estimate) for logistic regression is a special case of Newton's algorithm. If the problem is written in vector matrix form, with parameters $w^T = [\beta_0, \beta_1, \beta_2, \ldots]$, explanatory variables $x(i) = [1, x_1(i), x_2(i), \ldots]^T$ and expected value of Bernoulli distribution $\mu(i) = \dfrac{1}{1 + e^{-\mathbf{w}^T\mathbf{x}(i)}}$, the parameters $\mathbf{w}$ can be found using the following iterative algorithm:

$$\mathbf{w}_{k+1} = w_k - \alpha(\mathbf{X}^T\mathbf{B}_k\mathbf{X})^{-1}\mathbf{X}^T(\mu_{\mathbf{k}} - \mathbf{y})$$

where $\alpha$ is the learning rate, $\mathbf{B} = \mathrm{diag}(\mu(i)(1 - \mu(i)))$ is a diagonal weighted matrix, $\boldsymbol{\mu} = [\mu(1), \mu(2), \ldots]$ is the vector of expected values,

$$\mathbf{X} = \begin{bmatrix} 1 & x_1(1) & x_2(1) & \ldots \\ 1 & x_1(2) & x_2(2) & \ldots \\ \vdots & \vdots & \vdots & \end{bmatrix}$$

is the dataset in matrix form, and $\mathbf{y}(i) = [y(1), y(2), \ldots]^T$ is the vector of response variables.

It can be observed that the $\mathbf{B}$ matrix is of dimensionality $N \times N$, where $N$ is the number of instances. For large volume of data, this matrix can greatly affect the performance of the implementation. However, we need to only keep the values along the diagonal as this is a diagonal matrix; thereby precluding the potential performance drawbacks. The pseudo-code of both glm and our implementation are given in Algorithm 1 and 2 respectively. For each k-mer, we perform a hypothesis test to compute a p-value. The null hypothesis, i.e. the k-mer is not associated with the phenotype, is represented by fitting a logistic regression model against population structure and other confounders whereas for the alternate hypothesis we fit a logistic regression model against the confounders as well as the k-mer counts. Then a p-value is computed using a likelihood ratio test to check whether the null can be rejected.

In our implementation, there are two hyper parameters that need to be tuned before running the process. One is the learning rate of the logistic regression model and the other is the number of maximum iterations allowed for convergence. We used maximum iteration as 25

because we found that the glm implementation of R has 25 maximum iteration by default [13]. We used the learning rate value of 0.1.

**Algorithm 1**: glm

```
Inputs: X, y, α, MaxIter
Result: Weight vector w
Initialize w₀
k := 0
while k < MaxIter do
  Compute μₖ using wₖ;
  Compute Bₖ = diag(μ(i)(1 - μ(i))) using μₖ;
  Compute Error using y;
  if Error < ε then
    break;
  wₖ₊₁ := wₖ - α(Xᵀ Bₖ X)⁻¹ Xᵀ(μₖ - y);
  k := k + 1;
return w
```

**Algorithm 2**: HAWK

```
Inputs:
A (Phenotype of each individual)
B (Count of each k-mers in each individuals)
Z (Principal Components)
total (Total number of k-mers)
α := 0.1
MaxIter := 25
Compute y (yᵢ ∈ {0, 1}) using A
X_null := {Z, total}
Model_null := glm(X_null, y, α, MaxIter)
foreach k-mer kᵢ ∈ B do
  Compute the proportion of kᵢ, countᵢ, in each individual using B
  X_alt := {Z, countᵢ, total}
  Model_alt := glm(X_alt, y, α, MaxIter)
  Compute Likelihood_null of A using Model_null
  Compute Likelihood_alt of A using Model_alt
  Λ := Likelihood_alt / Likelihood_null
  p := chisq(2lnΛ, 1)
```

The implementation also makes it easier for users to specify the number of principal components to be used for population structure correction as well as additional covariates using command line parameters and input files.

## Bug fixes

An error was found in the old implementation regarding the order of samples during adjustment of p-values using the confounding factors. This has been corrected in the new C++ implementation.

## Implementation of Benjamini–Hochberg procedure

A number of approaches exist for adjusting p-value thresholds when multiple tests are being performed. Two such methods are: Bonferroni correction and Benjamini-Hochberg correction. The previous implementation performed hypothesis testing on each k-mer and performed Bonferroni correction using the total number of k-mers to determine k-mers associated with the phenotype in question. However, Bonferroni correction is known to be conservative i.e. it may fail to reject the null hypothesis even when it should be rejected [10].

Here, we implemented Benjamini-Hochberg correction which controls the false discovery rate (FDR). This can be used in studies underpowered for Bonferroni correction. The new implementations gives the provision for performing either correction according to the user preference.

## Support for Jellyfish 2

The original implementation of Hawk used a modified version of Jellyfish [7]. Subsequently, Jellyfish2 has been released which provides better performance. The present implementation of Hawk allows k-mer counting using a modified version of Jellyfish2 available through our Github repository.

## Re-implementation of post-processing

Once sequences corresponding to each loci associated with the phenotype are obtained, information such as average p-values of constituent k-mers as well as the average number of times they are present in case and control samples could be looked up using scripts provided with the original implementation. However, these scripts used a combination of C++ codes and shell commands, and was found to be slow in some cases [4]. Here, we re-implemented the script using hash tables in C++ to speed up the look-up.

# Results

To assess the performance and accuracy of our implementation, we use the *E.coli* dataset on ampicillin resistance which was analyzed using the original implementation of Hawk [5].

## Experimental setup

All the experiments are performed on a machine with CPU Intel(R) Xeon(R) CPU E5-2697 v2 @ 2.70GHz, 386GB memory with OS Ubuntu 18.04.3. There are two CPUs in the system with total 48 logical cores. The C++ implementation are compiled with g++-7.4.0 with no optimization flag. Execution time is measured using the "date" command.

## Hyper parameters

The code has 3 hyper parameters which are described below along with the values used in Table 1.

## Comparison of p-values

We have compared the output of our implementation and the output of the previous implementation by plotting logarithm of p-values obtained using the C++ implementation against that of p-values obtained using the R implementation in Fig 2. An identical result should give a line with slope 1. Graph produced by our implementation are almost linear with small deviations at a few points.

**Table 1. Hyper parameter values used in the implementation.**

| Hyper Parameter Name | Description | Value |
|---|---|---|
| learning_rate | Learning rate of the logistic model fitted in the code | 0.1 |
| max_iter | Maximum number of iterations the logistic model uses | 25 |
| CHUNK_SIZE | Number of samples the model reads at a time while fitting the model | 10000 |

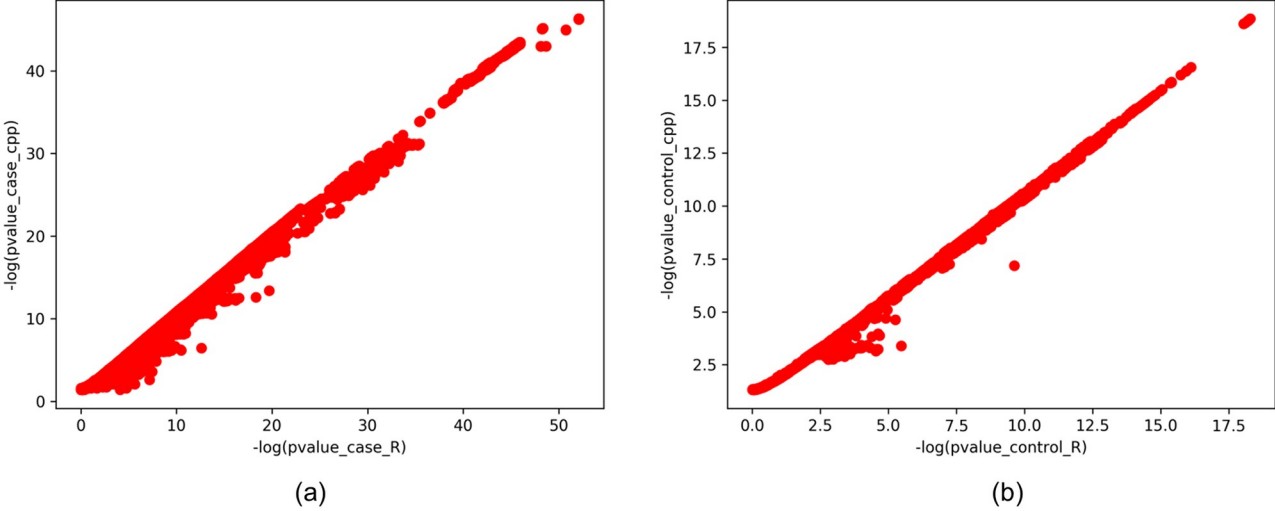

**Fig 2. Comparison of log of p-values with values computed using the previous implementation in R (-log(pvalue_case_R) and -log (pvalue_control_R)) along the x-axis and the ones computed using the new implementation using C++ (-log(pvalue_case_cpp) and -log (pvalue_control_cpp)) along the y-axis, for k-mers positively correlated with (a) cases, and (b) controls.**

The k-mers found significant after Bonferroni correction with threshold 0.05/176, 284, $643 = 2.84 \times 10^{-10}$ using the old implementation (corrected version) were mapped to *Escherichia coli* strain DTU-1 genome [GenBank: CP026612.1] and *Escherichia coli* strain KBN10P04869 plasmid pKBN10P04869A sequence [GenBank: CP026474.1]. The positions in the reference genomes and the p-values are shown in Manhattan plots in Fig 3(a) and 3(b). We find associations near the *β-lactamase TEM-1 (blaTEM-1)* gene, the presence of which is known to confer ampicillin resistance, as in [5]. However, some of the associations outside of this gene detected previously in [5], that are likely to be spurious, is no longer observed after the correction of error.

The above analysis was also performed with the k-mers found significant using the new C++ implementation and the Manhattan plots are shown in Fig 3(c) and 3(d). We observe that associations are detected in same regions as those found using the R implementation.

## Controlling FDR using the Benjamini-Hochberg procedure

HAWK uses Bonferroni correction to address multiple testing by default. However, we provide the option to control false discovery rate (FDR) using the Benjamini-Hochberg procedure. Fig 3(e) and 3(f) show Manhattan plots when k-mers are considered associated with ampicillin resistance after controlling FDR at level $\alpha = 0.05$ using the Benjamini-Hochberg procedure. We observe that many k-mers outside of the *β-lactamase TEM-1* gene are considered significant. We therefore recommend using Bonferroni correction and using the Benjamini-Hochberg procedure only if the study is under-powered for Bonferroni correction.

## Comparison of running time

The execution times of the old and new implementations are compared in Fig 4. Reported times are obtained by running both implementations using 32 threads. We find that the C++ implementation of the confounder correction phase is approximately three times faster than the previous implementation in R. However, the overall execution time is dominated by the k-mer counting step.

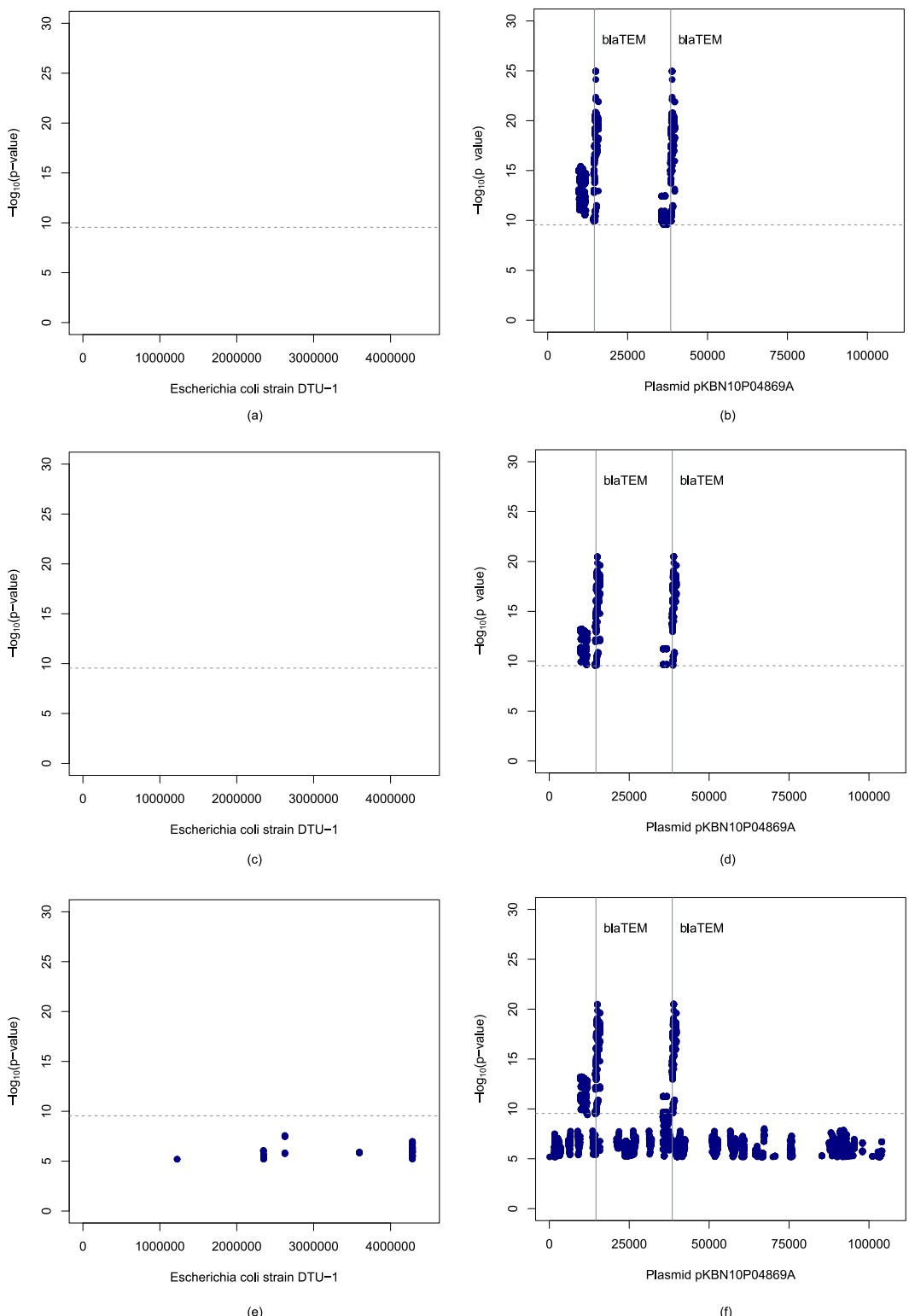

**Fig 3. Manhattan plots showing negative logarithms of adjusted p-values of k-mers found significantly associated with ampicillin resistance against their start positions in *Escherichia coli* strain DTU-1 genome (a), (c), (e), and plasmid pKBN10P04869A sequence (b), (d), (f); computed using the R implementation with Bonferroni correction (a), (b); using the C++ implementation with Bonferroni correction (c), (d); and using the C++ implementation with Benjamini-Hochberg correction with FDR level 0.05 (e), (f).** The horizontal lines indicate the threshold for Bonferroni correction at

$\frac{0.05}{176,284,643} = 2.84 \times 10^{-10}$ and the vertical lines denote start positions of *β-lactamase TEM-1* gene, the presence of which is known to confer resistance to ampicillin.

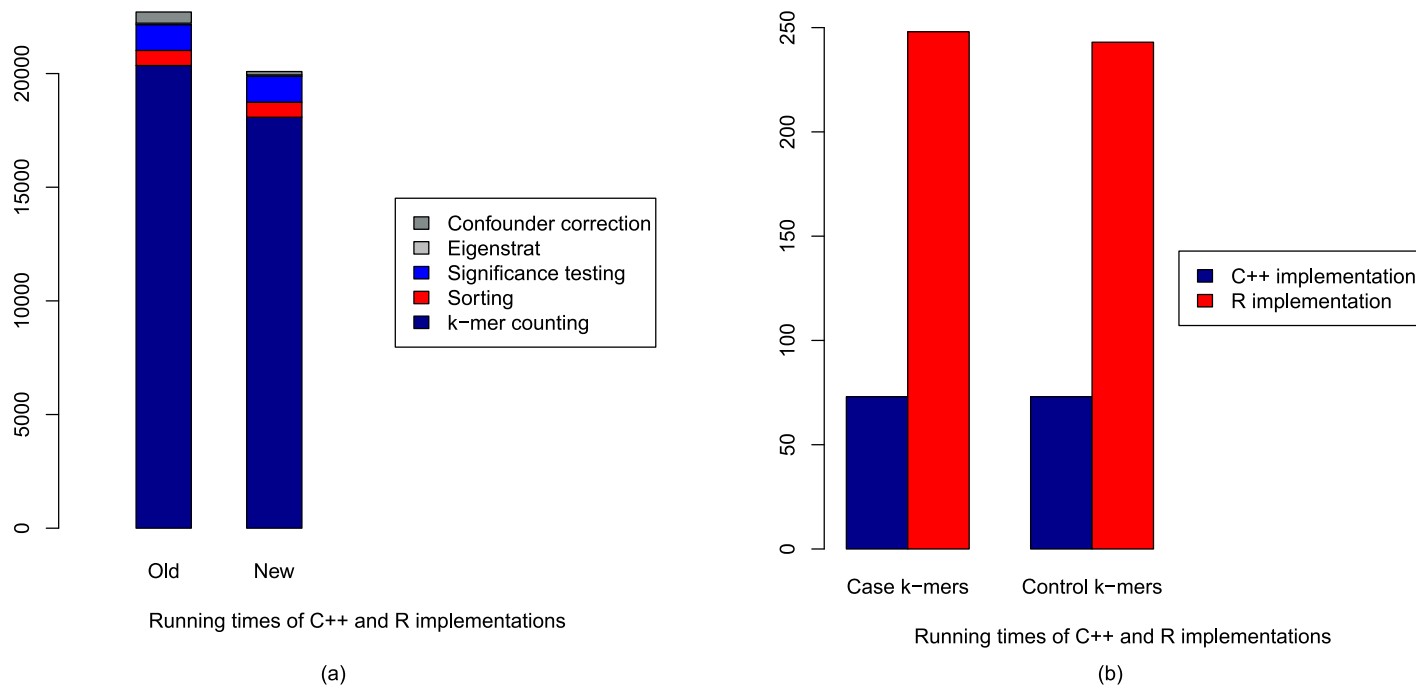

(a)

(b)

**Fig 4. Comparison of execution times of old and new implementations (a) for the entire pipelines, and (b) correction of confounding factors using C++ and R implementations.**

Table 2 shows comparison of execution times of Jellyfish and Jellyfish 2. We observe that, although Jellyfish 2 is faster overall, the performance improvement it provides is not substantial.

## Finding sex-specific sequences

The HAWK pipeline can be used to find sequences in sex chromosomes in organisms with unassembled or poorly assembled genomes. To assess the performance, we ran HAWK on sequencing data from the Yoruba in Ibadan, Nigeria (YRI) and the Toscani in Italia (TSI) populations from the 1000 genomes project dataset [11]. Of the 110 YRI and 109 TSI individuals, 107 were male and 112 were female. The sexes of individuals were used as cases and controls and the pipeline was executed.

The initial step revealed 106,272,845 and 17,056,781 k-mers that are present significantly more times in female and male samples respectively compared to the other. Principal

**Table 2. Comparison of running times of Jellyfish and Jellyfish 2.**

| Sub-command | Jellyfish (sec) | Jellyfish 2 (sec) |
|:---:|:---:|:---:|
| histo | 20 | 674 |
| count | 19,852 | 16,580 |
| dump | 481 | 824 |
| Total | 20,353 | 18,078 |

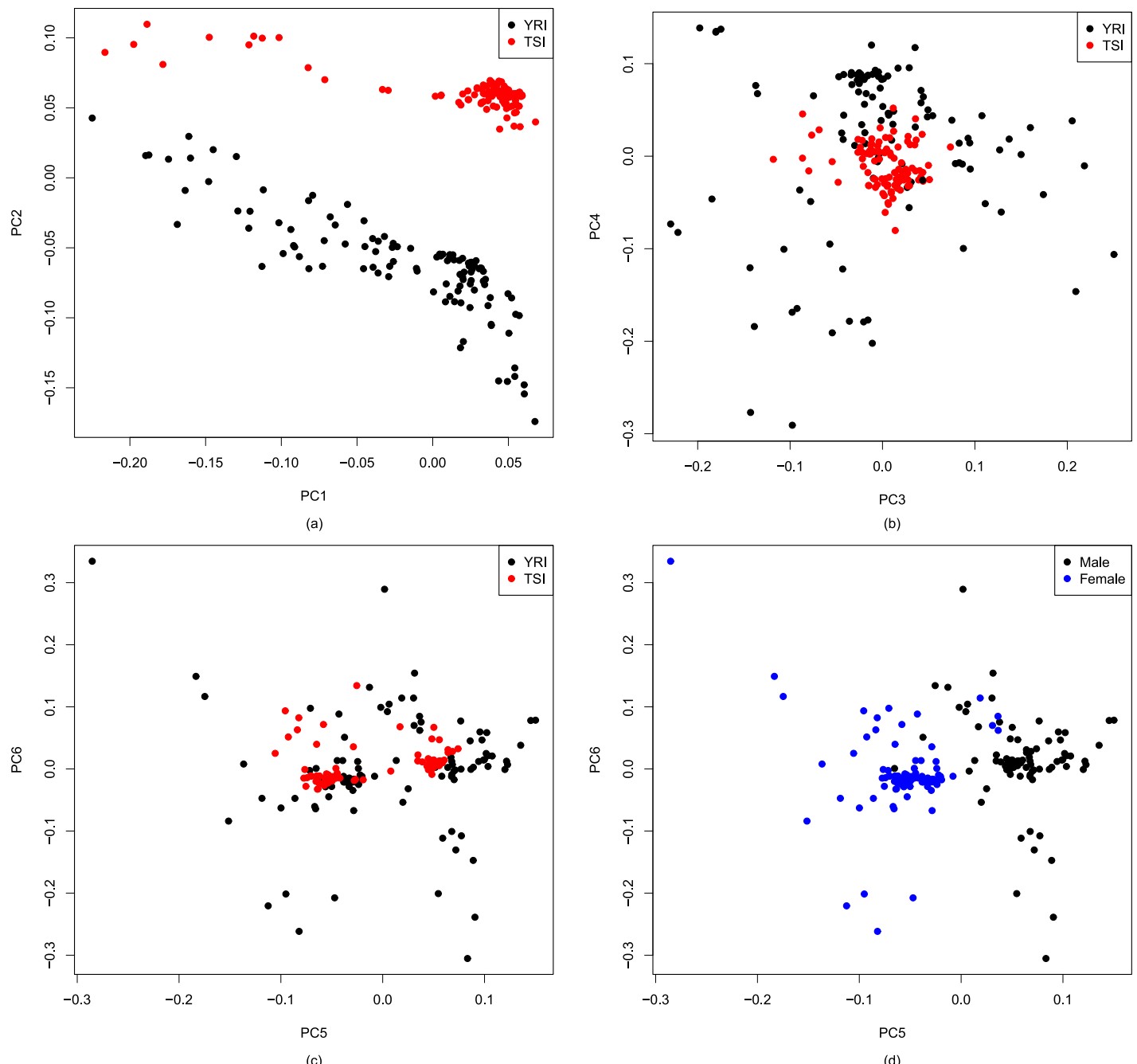

**Fig 5.** Principal component analysis (PCA) plots of the samples in the space formed by (a) first and second PCs, (b) third and fourth PCs, and (c) fifth and sixth PCs with samples colored according to populations. (d) PCA plots of the samples in the space formed fifth and sixth PCs with samples colored according to sex.

component analysis was then performed on the binary matrix denoting presence or absence of 32,699,548 randomly chosen k-mers, where each k-mer present in between 1% and 99% of the samples was selected with probability 0.01.

The PCA plots for the first six principal components are shown in Fig 5. The first six PCs explain 14.90% of the variance among which the first and second PCs explain 7.68% and 2.51% of the total variance respectively. We observe that the first two PCs capture the

**Table 3. Summary of k-mers found positively correlated with female and male samples.**

| Sex | Total | Chr X | Chr Y | Others | Unmapped |
|---|---|---|---|---|---|
| Female | 54,256,206 | 54,241,253 | 148 | 1,474 | 13,331 |
| Male | 14,473,058 | 6,197 | 13,947,961 | 63,454 | 455,446 |

population structure whereas no relationship is seen between populations and the next four PCs. Hence PC1 and PC2 are used as confounders, as done by default. It may be noted from Fig 5(d) that the fifth PC nearly separate the two sexes. As such treating this as a confounder would lead to removal of many sex-specific k-mers.

After correcting for confounders, we obtain 14,473,058 and 54,256,206 k-mers positively correlated with male and female samples respectively. The k-mers were mapped to the human reference genome (hg38/GRCh38 [GCA_000001405.15]) using Bowtie 2 [14] to analyze their locations. The results are summarized in Table 3. We find that 99.97% and 96.37% of k-mers positively correlated with female and males samples map to Chromosome X and Chromosome Y respectively. The remaining k-mers map to other locations in the human reference genome or stay unmapped.

The positions of the k-mers, positively correlated with female and male samples, in Chromosomes X and Y respectively are shown in Fig 6(a) and 6(b). We observe that k-mers throughout the entire sequenced regions of the two chromosomes are detected using HAWK. It is worth noting that the region in Chromosome Y, where no k-mer could be mapped, is missing from the reference genome i.e. represented by a sequence of Ns (the wild card character used to denote any nucleic acid in fasta format) in the reference [15]. Our analysis reveals that the k-mers that can be mapped to Chromosome Y have an average total count of 754.10 in the male samples whereas that of the unmapped k-mers is 2730.06. The histograms of counts of mapped and unmapped k-mers in Fig 6(c) also show that the count distribution of unmapped k-mers has a heavier tail compared to that of the mapped ones. This suggests that many of the k-mers positively correlated to male samples that could not be mapped to Chromosome Y are from the missing region in the chromosome since the missing region is known to be rich in repeats.

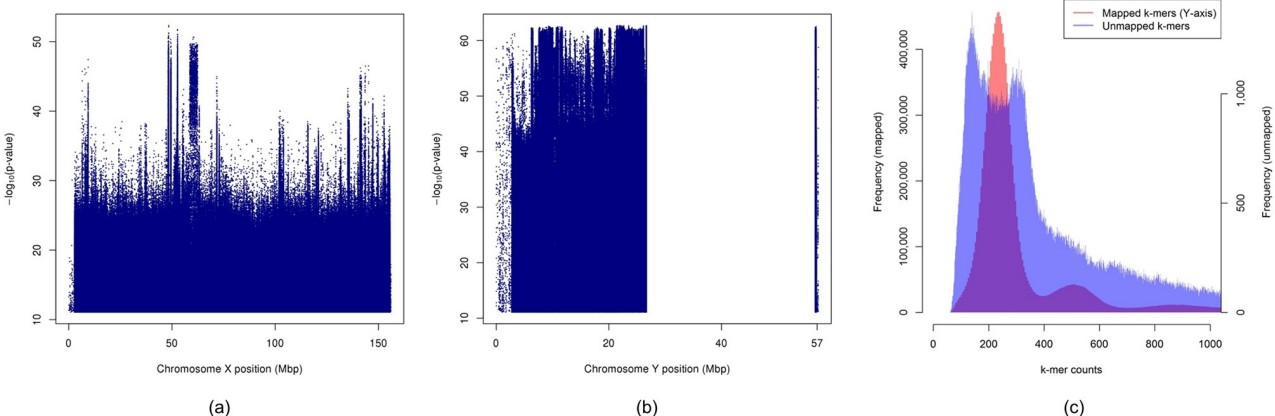

(a) (b) (c)

**Fig 6.** Manhattan plots showing negative logarithm of p-values of (a) k-mers positively correlated with female samples against their positions in Chromosome X, and (b) k-mers positively correlated with male samples against their positions in Chromosome Y. The region in Chromosome Y, where no k-mer mapped to, is missing from the reference genome i.e. represented by a sequence of Ns in the reference. (c) Histograms of counts of k-mers positively correlated with male samples that were mapped to Chromosome Y and those which could not be mapped.

## Discussion

We have re-implemented portions of Hawk, which is a tool for association mapping using k-mers. The re-implementation in C++ makes it faster and more convenient to use while retaining accuracy. We have also added support for the new version of k-mer counting tool Jellyfish and correction for multiple testing using the Benjamini-Hochberg procedure. The k-mer counting step remains the bottleneck in the pipeline which may be addressed by when faster k-mer counting tools emerge. Finally, we show how the method can be applied to determine sex specific sequences in organisms accurately.

## Acknowledgments

We thank Guillaume Marçais for providing the patch to modify Jellyfish 2 according to the requirements of HAWK. We also thank Lior Pachter for providing helpful comments and resources during the project.

## Author Contributions

**Conceptualization:** Atif Rahman.

**Formal analysis:** Zakaria Mehrab, Jaiaid Mobin, Ibrahim Asadullah Tahmid, Atif Rahman.

**Methodology:** Zakaria Mehrab, Jaiaid Mobin, Atif Rahman.

**Project administration:** Atif Rahman.

**Software:** Zakaria Mehrab, Jaiaid Mobin.

**Supervision:** Atif Rahman.

**Validation:** Zakaria Mehrab, Jaiaid Mobin, Ibrahim Asadullah Tahmid, Atif Rahman.

**Visualization:** Zakaria Mehrab, Jaiaid Mobin, Ibrahim Asadullah Tahmid, Atif Rahman.

**Writing – original draft:** Zakaria Mehrab, Jaiaid Mobin, Ibrahim Asadullah Tahmid, Atif Rahman.

**Writing – review & editing:** Zakaria Mehrab, Jaiaid Mobin, Ibrahim Asadullah Tahmid, Atif Rahman.

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
