## [Decision Letter · Decision Letter 0]

11 Sep 2020

PONE-D-20-20743

Efficient association mapping from k-mers - an application in finding sex-specific sequences

PLOS ONE

Dear Dr.Rahman:,

Thank you for submitting your manuscript to PLOS ONE. After careful consideration, we feel that it has merit but does not fully meet PLOS ONE’s publication criteria as it currently stands. Therefore, we invite you to submit a revised version of the manuscript that addresses the points raised during the review process.

We look forward to receiving your revised manuscript.

Kind regards,

Momiao Xiong

Academic Editor

PLOS ONE

Journal Requirements:

Reviewers' comments:

Reviewer's Responses to Questions

**Comments to the Author**

1. Is the manuscript technically sound, and do the data support the conclusions?

Reviewer #1: Yes

Reviewer #2: Yes

2. Has the statistical analysis been performed appropriately and rigorously? 

Reviewer #1: Yes

Reviewer #2: Yes

3. Have the authors made all data underlying the findings in their manuscript fully available?

Reviewer #1: Yes

Reviewer #2: No

4. Is the manuscript presented in an intelligible fashion and written in standard English?

Reviewer #1: Yes

Reviewer #2: Yes

5. Review Comments to the Author

Reviewer #1: The manuscript “Efficient association mapping from k-mers - an application in finding sex-specific sequences” reimplemented a previously developed algorithm HAWK using C++ instead of the original R scripts and/or other scripts. Briefly, the major changes include the reimplementation of association running and the multiple test correction. The package was tested in comparison with the original HAWK package. Both results are similar on the E. coli data. The authors also ran the new package to identify sequences of sex chromosomes using human population whole genome sequencing data. It showed that significant k-mers are largely from chromosomes X and Y. Running time was also compared. Overall the manuscript aims to computationally improve the HAWK performance for k-mer GWAS. The computational speed for k-mer GWAS is important because hundreds of million statistical tests could be needed for a k-mer GWAS study. Below I have a few minor comments.

1. The lab for x- and y-axis could be clearer, although the audience could guess what cpp means and what R means. Notes can be added in the legend.

2. The paragraph for the explanation of k-mer missing regions on chromosome Y is hard to understand. If the region is missed in the reference genome as the authors speculated, what was plotted in Fig 5b? And it is not reasonable to state that “The missing region in Chromosome Y also possibly explains the large percentage of male associated k-mers that could not be mapped to it in comparison to the percentage of female associated k-mers that map to Chromosome X.” because the difference in the mapping percentage between male and female significance k-mers is not large (99.57 vs 96.37%) but the regions seem to be large. An alternative explanation is that X and Y share sequences and k-mers derived from the common sequences would not be identified as male associated k-mers, the term used by the authors.

3. Female and male associated k-mers were used in the manuscript. Actually, both k-mers sets are associated with the gender (male and female). The authors probably categorized them based on the direction (positive/negative) of correlation. I think, more accurate terms are needed to be used.

4. Line9-11 in the Introduction are not clear. What kind of method it is and why the method made the reference-based method hard.

Reviewer #2: The authors improved the implementation of the Hawk, an alignment-free association mapping method for genetic research, by reducing the running time and addition of feature of Benjamini-Hochberg method for controlling for multiple comparisons. The authors also suggest that the method can be used to identify sex-specific sequences. This study has several important strengths, however, there are several issues that should be clarified and some details should be added before publication.

1. Introduction: I think two flow-charts, one for the original HAWR, and the other for re-implemented one by the authors, would be helpful that readers can understand the differences in procedures more intuitively.

2. Implementation, page 2: Please add more details about the population structure. Do they include multiple variables or is it an integrated one? How can it be used for controlling for confounding, for example, by adjusting in the regression model?

3. Implementation, page 2, line 50-52: “p-values were adjusted for population structure…” This sentence is unclear. Does this mean the associations between genes and phenotypes were adjusted for confounders and p-values were estimated, or the p-values themselves were corrected? If the former is the case, then it should be said that the p-values were adjusted.

4. Results, Figure 1: Please indicate which plot is for the original or revised one.

5. Results, FDR: What the FDR value and corresponding alpha value were used for example in Figure 2(e) and (f)? It will be also more informative if the author could show the alpha level from the Bonferroni correction in the same example.

6. Results, sex-specific sequences: How the number of 32,699,548 of k-mers was chosen for the PCA?

7. Results, sex-specific sequences: How much variance can be explained by PC1 to PC6?

8. Results, sex-specific sequences: It would be more informative to plot PC1 to PC6 by populations. If PC5 and PC6 are also significantly different between YRI and TSI, a senstivity analysis with additional adjustment for PC5 and PC6 would be necessary.

6. PLOS authors have the option to publish the peer review history of their article (what does this mean?). If published, this will include your full peer review and any attached files.

Reviewer #1: No

Reviewer #2: No

---

## [Author Response · Author response to Decision Letter 0]

28 Oct 2020

Reviewer 1

The manuscript “Efficient association mapping from k-mers - an application in finding sex-specific sequences” reimplemented a previously developed algorithm HAWK using C++ instead of the original R scripts and/or other scripts. Briefly, the major changes include the reimplementation of association running and the multiple test correction. The package was tested in comparison with the original HAWK package. Both results are similar on the E. coli data. The authors also ran the new package to identify sequences of sex chromosomes using human population whole genome sequencing data. It showed that significant k-mers are largely from chromosomes X and Y. Running time was also compared. Overall the manuscript aims to computationally improve the HAWK performance for k-mer GWAS. The computational speed for k-mer GWAS is important because hundreds of million statistical tests could be needed for a k-mer GWAS study. Below I have a few minor comments.

Comment 1

The lab for x- and y-axis could be clearer, although the audience could guess what cpp means and what R means. Notes can be added in the legend.

Response

We have clarified the labels for x- and y-axis in the caption of Figure 2 on Page 6.

Comment 2

The paragraph for the explanation of k-mer missing regions on chromosome Y is hard to understand. If the region is missed in the reference genome as the authors speculated, what was plotted in Fig 5b? And it is not reasonable to state that “The missing region in Chromosome Y also possibly explains the large percentage of male associated k-mers that could not be mapped to it in comparison to the percentage of female associated k-mers that map to Chromosome X.” because the difference in the mapping percentage between male and female significance k-mers is not large (99.57 vs 96.37\\%) but the regions seem to be large. An alternative explanation is that X and Y share sequences and k-mers derived from the common sequences would not be identified as male associated k-mers, the term used by the authors.

Response

There is a sequence of Ns that corresponds to the missing region in the reference fasta file (Lines 205-207). Hence, although nothing could be mapped to the region, the Manhattan plot could be generated.

We have rewritten this portion (Lines 207-214) which should hopefully make it more reasonable. We have added a figure (Figure 6(c)) showing the histograms of counts of k-mers that could and could not be mapped to Chromosome Y. The heavier tail of distribution of unmapped k-mer counts indicate that they are from a repeat rich region which is what makes the missing region hard to assemble.

We agree with the alternate explanation in the sense that if the difference in k-mer count distributions in cases and controls is small and the number of samples in the study is not large, some k-mers may not be identified. However, this is unlikely in this case because (a) we are able to identify k-mers from throughout Chromosome X, and (b) our earlier analysis indicate (not shown here) that it is easier to identify the k-mers that have one copy in one group and none in the other (Chromosome Y) than the k-mers with two copies in one group and one copy in the other group (Chromosome X). 

Comment 3

Female and male associated k-mers were used in the manuscript. Actually, both k-mers sets are associated with the gender (male and female). The authors probably categorized them based on the direction (positive/negative) of correlation. I think, more accurate terms are needed to be used.

Response

We thank the reviewer for the observation. We have corrected the terminology.

Comment 4

Line9-11 in the Introduction are not clear. What kind of method it is and why the method made the reference-based method hard.

Response

We have rewritten the lines to make them more comprehensible. (Lines 9-20)

Reviewer 2

The authors improved the implementation of the Hawk, an alignment-free association mapping method for genetic research, by reducing the running time and addition of feature of Benjamini-Hochberg method for controlling for multiple comparisons. The authors also suggest that the method can be used to identify sex-specific sequences. This study has several important strengths, however, there are several issues that should be clarified and some details should be added before publication.

Comment 1

Introduction: I think two flow-charts, one for the original HAWR, and the other for re-implemented one by the authors, would be helpful that readers can understand the differences in procedures more intuitively.

Response

We thank the reviewer for suggestion. We have added a flowchart (Figure 1) showing old features and highlighting re-implementations.

Comment 2

Implementation, page 2: Please add more details about the population structure. Do they include multiple variables or is it an integrated one? How can it be used for controlling for confounding, for example, by adjusting in the regression model?

Response

We have added more details about the population structure (Lines 56-60) and the regression model (Lines 84-88).

Comment 3

Implementation, page 2, line 50-52: “p-values were adjusted for population structure…” This sentence is unclear. Does this mean the associations between genes and phenotypes were adjusted for confounders and p-values were estimated, or the p-values themselves were corrected? If the former is the case, then it should be said that the p-values were adjusted.

Response

It is indeed the former. We have rewritten the sentence to make it clearer. (Lines 61-63)

Comment 4

Results, Figure 1: Please indicate which plot is for the original or revised one.

Response

We have modified the caption of Figure 1 to make this clear.

Comment 5

Results, FDR: What the FDR value and corresponding alpha value were used for example in Figure 2(e) and (f)? It will be also more informative if the author could show the alpha level from the Bonferroni correction in the same example.

Response

We use alpha level of 0.05. We have now mentioned this in the text (Line 161) and in the caption of Figure 3. We have also mentioned the alpha level for Bonferroni correction in the text (Line 145) and added horizontal lines in Figure 3 to highlight the threshold. 

Comment 6

Results, sex-specific sequences: How the number of 32,699,548 of k-mers was chosen for the PCA?

Response

Each k-mer present in between 1\\% and 99\\% of the samples are chosen with probability 0.01 which led to this number. Clarified this in the manuscript. (Lines 185-186)

Comment 7

Results, sex-specific sequences: How much variance can be explained by PC1 to PC6?

Response

The first six PCs explain 14.90\\% of the variance among which the first and second PCs explain 7.68\\% and 2.51\\% of the total variance respectively. We have now mentioned this in the paper. (Lines 187-189)

Comment 8

Results, sex-specific sequences: It would be more informative to plot PC1 to PC6 by populations. If PC5 and PC6 are also significantly different between YRI and TSI, a sensitivity analysis with additional adjustment for PC5 and PC6 would be necessary.

Response

We have added plots for PC3 to PC6 by populations in Figure 5. We had not observed any significant relation between populations and PC3-PC6.

---

## [Decision Letter · Decision Letter 1]

22 Dec 2020

Efficient association mapping from k-mers - an application in finding sex-specific sequences

PONE-D-20-20743R1

Dear Dr. Rahman,

We’re pleased to inform you that your manuscript has been judged scientifically suitable for publication and will be formally accepted for publication once it meets all outstanding technical requirements.

Kind regards,

Momiao Xiong

Academic Editor

PLOS ONE

Additional Editor Comments (optional):

Reviewers' comments:

Reviewer's Responses to Questions

**Comments to the Author**

1. If the authors have adequately addressed your comments raised in a previous round of review and you feel that this manuscript is now acceptable for publication, you may indicate that here to bypass the “Comments to the Author” section, enter your conflict of interest statement in the “Confidential to Editor” section, and submit your "Accept" recommendation.

Reviewer #1: All comments have been addressed

Reviewer #2: All comments have been addressed

2. Is the manuscript technically sound, and do the data support the conclusions?

Reviewer #1: (No Response)

Reviewer #2: Yes

3. Has the statistical analysis been performed appropriately and rigorously? 

Reviewer #1: Yes

Reviewer #2: Yes

4. Have the authors made all data underlying the findings in their manuscript fully available?

Reviewer #1: Yes

Reviewer #2: Yes

5. Is the manuscript presented in an intelligible fashion and written in standard English?

Reviewer #1: Yes

Reviewer #2: Yes

6. Review Comments to the Author

Reviewer #1: Authors have performed additional analyses to clarify some confusions. The revision has addressed reviewer's comments.

Reviewer #2: The authors addressed all my concerns. I have no further comments.

7. PLOS authors have the option to publish the peer review history of their article (what does this mean?). If published, this will include your full peer review and any attached files.

Reviewer #1: No

Reviewer #2: No

---

## [Editor Report · Acceptance letter]

29 Dec 2020

PONE-D-20-20743R1 

Efficient association mapping from k-mers - an application in finding sex-specific sequences 

Dear Dr. Rahman:

I'm pleased to inform you that your manuscript has been deemed suitable for publication in PLOS ONE. Congratulations! Your manuscript is now with our production department. 

Kind regards, 

on behalf of

Prof. Momiao Xiong 

Academic Editor

PLOS ONE